https://doi.org/10.1038/s42004-022-00789-y · OPEN
# Ion-specific binding of cations to the carboxylate and of anions to the amide of alanylalanine

Carola Sophie Krevert [1], Lucas Gunkel [1], Constantin Haese[2] & Johannes Hunger [1✉]

Studies of ion-specific effects on oligopeptides have aided our understanding of Hofmeister effects on proteins, yet the use of different model peptides and different experimental sensitivities have led to conflicting conclusions. To resolve these controversies, we study a small model peptide, L-Alanyl-L-alanine (2Ala), carrying all fundamental chemical protein motifs: C-terminus, amide bond, and N-terminus. We elucidate the effect of GdmCl, LiCl, KCl, KI, and KSCN by combining dielectric relaxation, nuclear magnetic resonance ([1]H-NMR), and (two-dimensional) infrared spectroscopy. Our dielectric results show that all ions reduce the rotational mobility of 2Ala, yet the magnitude of the reduction is larger for denaturing cations than for anions. The NMR chemical shifts of the amide group are particularly sensitive to denaturing anions, indicative of anion-amide interactions. Infrared experiments reveal that LiCl alters the spectral homogeneity and dynamics of the carboxylate, but not the amide group. Interaction of LiCl with the negatively charged pole of 2Ala, the $COO^-$ group, can explain the marked cationic effect on dipolar rotation, while interaction of anions between the poles, at the amide, only weakly perturbs dipolar dynamics. As such, our results provide a unifying view on ions' preferential interaction sites at 2Ala and help rationalize Hofmeister effects on proteins.

[1] Department of Molecular Spectroscopy, Max Planck Institute for Polymer Research, Ackermannweg 10, 55128 Mainz, Germany. [2] Department of Molecular Electronics, Max Planck Institute for Polymer Research, Ackermannweg 10, 55128 Mainz, Germany. ✉email: hunger@mpip-mainz.mpg.de

Since the discovery by Hofmeister that salts can specifically (de-)stabilize proteins in solution[1], and thereby alter protein functions[2–5], specific ion effects have been intensively studied[6–16]. In recent years, there has been increasing evidence for such specific ion effects originating from direct ion–protein interactions[17–20]. Yet, the manifold of potential protein interaction sites—including charged residues, hydrophobic fragments, and the amide backbone—make it difficult to resolve the site-specificity of Hofmeister effects. Moreover, not only the chemical nature of the protein sites but also their hydration—intimately connected to the protein structure—affects interaction with ions[21]. Together, this complexity makes it difficult to rationalize specific ion effects on a molecular level[11,20,22,23].

Model systems, like amide-rich polymers, have tremendously helped understanding the underlying principles of specific ion effects on proteins[24–26], as the effect of salts on the phase transition of the polymers resembles the effect on proteins. Ion-specific macroscopically observable phase transitions have been traced back to microscopic interactions of ions with the amide backbone: spectroscopic studies and molecular dynamics simulations have confirmed direct ion-amide interactions[13,27–31]. In addition, the hydration of the macromolecule can affect ion binding[21], and interactions between anions and cations in bulk compete with ion-amide binding[25,32,33]. Yet, considering only the interaction of ions with amide groups is insufficient to explain the salts' impact on protein stabilities[13,23]. Clearly, interaction sites beyond the amide backbone have to be taken into account[18,23]. Indeed, the charged termini have been shown to markedly affect the solubility of model peptides[34] and interactions with charged residues have been suggested to be critical, particularly for the strongly denaturing guanidinium cation[16,18,35]. As such, to fully understand specific ion effects on proteins, peptides, and polymer model systems necessitates going beyond model system containing an individual site (e.g., amide) and considering multiple binding sites and competitive interactions[18].

The above conclusions have been drawn using a variety of model compounds, including simple molecules like N-methyl-acetamide[13,36–38], short oligopeptides[16,39,40], and polymers[21,24,38,41,42]. Comparison of the results using small molecules to those obtained with macromolecules[21,38] has suggested that small model systems cannot account for all details of ion-specific effects on proteins and the length of the macromolecule largely impacts the interaction with ions. This different behavior has been explained by the different hydration structures of small molecules and macromolecules[21]. Conversely, using large, conformationally flexible molecules makes it more challenging to isolate ion interactions: an ion-induced change of an experimental observable may stem from direct interaction of an ion with the model system or from an indirect, ion-induced change in the conformation[16]. For instance, hydration of salt may lead to dehydration and conformational variation of a macromolecule[43].

In addition to the complexity due to the use of different molecular models with different conformational degrees of freedom, also the use of different experimental approaches with different sensitivities adds an experimental complexity. So far, different experimental techniques have been used to elucidate ions' interactions with protein sites. With dielectric relaxation (DR) spectroscopy, the reduction of the dipolar rotational degree of freedom of model peptides can be readily detected[14,15,44]. As this reduction is considered to stem from coupling of the ions' translation to the peptides' dipolar rotation, the proximity of both, cations and anions to the peptide can be detected with equal sensitivity, but site-specific information cannot be obtained[13–16]. Site-specific information can be readily obtained from nuclear magnetic resonance (NMR) experiments. The salt-induced variation of the chemical shift of peptide nuclei has been demonstrated to sensitively report on the site-specific interaction of ions (predominantly anions) with the peptides' protons[18,31,34,39]. Information on the interaction with cations is more challenging to extract from NMR experiments[37,45]. In turn, vibrational spectroscopies, like infrared absorption (IR) spectroscopy, can detect (cat)ion-induced changes in the environment of the amide carbonyl group or of carboxylate groups[46,47]. Yet, monovalent cations hardly affect these vibrations[27] and marked shifts in the vibrational frequencies are only observed for bivalent cations[27,48]. As such, orthogonal conclusions have been obtained with either cations[13,29,49] or anions[11,31,38] interacting more strongly with the model peptides.

These seemingly contradicting conclusions may be partly explained by different experimental sensitivities: linear vibrational spectroscopies such as infrared absorption spectroscopy or Raman spectroscopy show little sensitivity to interactions with monovalent cations[50]. However, the sensitivity can be enhanced by using nonlinear vibrational spectroscopies, as not only the shift of the resonance frequency can be detected, but also the dynamics with which the oscillators adapt their frequency as a response to fluctuations in their direct environment[46]. Using these fluctuations, salt-induced changes to the dynamic environment of the vibrational probes can be uncovered. Thus, combining nonlinear spectroscopies with previously used spectroscopies (DR, NMR, and IR) on the very same peptide has the potential to experimentally elucidate the salt-specific distribution of ions around model peptides in solution.

Here, we combine these experiments to study specific interactions of different ions around zwitterionic alanyl-alanine (2Ala) in solution. 2Ala carries all functional groups discussed above—both termini and an amide bond. Although 2Ala contains only one amide bond, and thus cannot fully represent the amide backbone of proteins, 2Ala has a rather rigid conformation[51], which limits changes in the spectroscopic observables due to conformational changes and thus allows isolating ionic interactions spectroscopically. To study specific ion effects on 2Ala, we chose two prominent cationic denaturants (GmdCl, LiCl), two salts with denaturing anions (KI, KSCN), and KCl as a reference. Our dielectric data show that the denaturing anions only moderately reduce the rotational mobility of 2Ala, and the NMR results suggest that these anions predominantly bind to the amide N–H group. GdmCl and LiCl have a much stronger effect on the rotation of 2Ala in solution. While none of the salts alters the infrared absorption spectra for the carboxylate and the amide CO groups significantly, 2D-IR experiments show that spectral dynamics of the COO⁻ group are slowed down in the presence of LiCl, indicative of Li⁺ interacting with the C-terminus. Together, this site-specific information and the deduced competitive binding provide a geometric rationale for apparently contradicting previous observations for ion-specific effects on model peptides.

## Results and discussion

To provide a consistent view on ion-specific effects on a small model peptide, we herein use a combination of experimental approaches on solutions of the same dipeptide, alanlylalanine, which probe different properties of 2Ala: salt-induced changes to the dynamics of its electrical dipole, site-specific changes to the chemical environment of individual protons, and ion-induced changes to the carbonyl/carboxylate vibrational structure and dynamics (Fig. 1). In the following, we will first discuss these experiments individually.

**Rotational mobility of 2Ala probed by dielectric spectroscopy.** To study the effect of ions on the rotational mobility of 2Ala we

use DR spectroscopy[16], which probes the polarization of a sample in an externally applied oscillating electric field with field frequency $\nu$. The polarization is typically expressed in terms of the complex permittivity spectrum ($\hat{\varepsilon} = \varepsilon'(\nu) - i\varepsilon''(\nu)$), with the real permittivity, $\varepsilon'$, corresponding to the in-phase polarization and the dielectric loss, $\varepsilon''$, representing the out-of-phase (absorptive) polarization. For liquids at Gigahertz frequencies, polarization typically stems from the rotation of dipolar molecules. Thus, besides dipolar water, the reorientation of highly dipolar, zwitterionic peptides gives rise to marked polarization contributions[52,53]. The orientational relaxation of each dipolar species is characterized by dispersion in $\varepsilon'$ and a loss peak at the rotational frequency of the dipolar species[54].

Before studying the effect of Hofmeister salts, we first characterize the polarization dynamics of aqueous 2Ala solutions. The dielectric spectrum of a 0.25 M 2Ala solution in water (Fig. 2a) exhibits two well-separated dispersions in $\varepsilon'(\nu)$ and two peaks in $\varepsilon''(\nu)$ at ~20 GHz and ~1 GHz, respectively, evidencing two disparate relaxations. The higher-frequency relaxation is due to the orientational relaxation of water's hydrogen-bonded network at ~20 GHz[55]. The lower-frequency relaxation can be ascribed to the rotational relaxation of zwitterionic 2Ala—similar to earlier findings for other oligopeptides or amino acids[15,16,56]. Consistent with this notion, the relaxation strength (loss peak amplitude) at ~1 GHz increases with increasing 2Ala concentration (Fig. 2b).

To exclude marked 2Ala–2Ala interactions (i.e., dipole–dipole correlations) contributing to the observed 2Ala relaxation, we quantify the contribution of both relaxations to the experimental spectra. In line with our earlier studies[13,14,16], we model the experimental spectra using a Debye mode[57] for the 2Ala

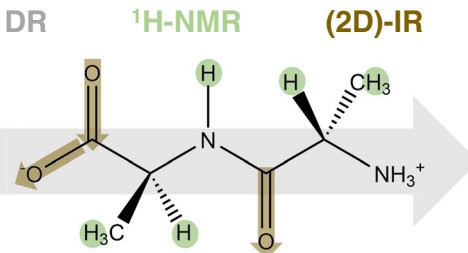

**Fig. 1 Schematic illustration of the combined experimental approach to study ion-specific effects on 2Ala.** Dielectric relaxation spectroscopy probes the dynamics of 2Ala's electrical dipole (gray arrow), $^1$H-NMR spectroscopy the chemical environment of 2Ala's protons (marked green), and 2D-IR detects the vibrational dynamics of the CO and anti-symmetric COO$^-$ stretching vibrations of 2Ala (marked brown).

contribution and a Cole–Cole[58,59] mode for the water relaxation:

$$\hat{\varepsilon}(\nu) = \frac{S_{2Ala}}{1 + (2\pi i \nu \tau_{2Ala})} + \frac{S_{H2O}}{1 + (2\pi i \nu \tau_{H2O})^{1-\alpha}} + \varepsilon_\infty + \frac{\kappa}{2\pi i \nu \varepsilon_0} \quad (1)$$

with the relaxation amplitude $S_j$, and the relaxation time $\tau_j$. The Cole–Cole parameter $\alpha$ accounts for the symmetrical broadening of the water relaxation as compared to a Debye relaxation. $\varepsilon_\infty$ accounts for all polarizations occurring at frequencies higher than covered in our experiment (e.g., electronic, vibrational contributions). The last term of Eq. (1) accounts for losses due to the finite conductivity of the samples, where we assume the conductivity $\kappa$ to be real and frequency-independent. $\varepsilon_0$ is the permittivity of free space.

The model in Eq. (1) describes the experimental spectra very well (Fig. 2a, b), and the obtained parameters are shown in the supporting information (see Supplementary Fig. 1 and Supplementary Discussion 1). Most importantly, the peptide's amplitude $S_{2Ala}$ increases linearly with increasing peptide concentration. For uncorrelated rotation of 2Ala, $S_{2Ala}$ is related to the 2Ala concentration and the squared effective dipole moment $\mu_{eff}$(2Ala)[16]. We find this dipole moment as calculated from $S_{2Ala}$ according to ref. [16] (see Supplementary Eq. (1)) to be constant $\mu_{eff}$(2Ala) $\approx$30 D (Fig. 2c). The insensitivity of $\mu_{eff}$(2Ala) to concentration demonstrates that 2Ala zwitterions do not exhibit preferred parallel or antiparallel dipolar correlations and do not undergo marked conformational changes at concentrations up to 250 mM. As such, salt-induced changes of $S_{2Ala}$ due to ion-induced changes in the conformation or screened dipolar correlations are rendered unlikely.

After having established that 2Ala's contribution to the dielectric response is due to uncorrelated rotational motion of the peptides' zwitterions, we now focus on how different salts affect 2Ala in solution. As can exemplarily be seen in Fig. 3a, the addition of LiCl to a $c_{2Ala} = 0.25$ M solution results in a marked depolarization—a reduction of the dielectric permittivity and loss—with increasing salt concentration (for other salts, see Supplementary Fig. 2). This depolarization stems in part from the dilution of water due to the salt, as the volume concentration of water decreases with increasing salt concentration[14]. In addition, kinetic depolarization[60,61] upon adding salt reduces the dipolar response: The dipolar molecules tend to align according to the local electric field of a translating ion, rather than to the externally applied electric field, which reduces the dipolar relaxation amplitude. As kinetic depolarization only relies on the coupling between the translation of ions and rotation of dipoles in an external electric field, the dipolar response of both, water and 2Ala, is subject to kinetic depolarization. Indeed, zooming into

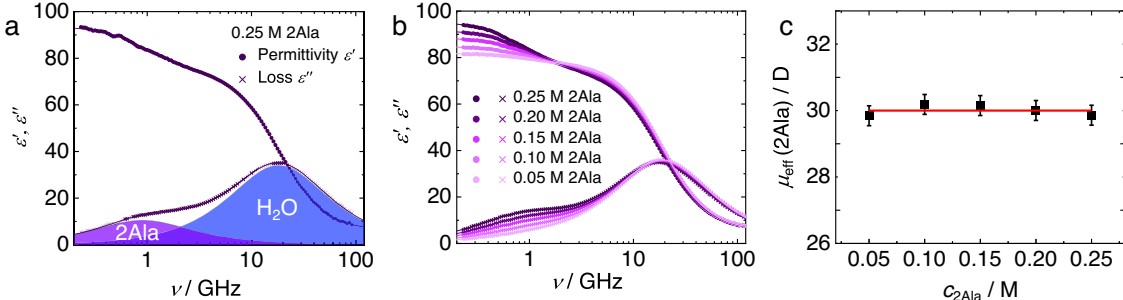

**Fig. 2 Dielectric spectra of aqueous 2Ala solutions are consistent with uncorrelated dipolar rotation of 2Ala.** Complex permittivity spectrum of (**a**) 0.25 M aqueous 2Ala solution and **b** aqueous 2Ala solutions with increasing concentration of 2Ala. Solid circles and crosses show the experimental dielectric permittivity ($\varepsilon'$) and dielectric loss spectra ($\varepsilon''$), respectively. Solid lines show fits of Eq. (1) to the experimental data. The contribution of the 2Ala and the water relaxations to the dielectric loss are shown as shaded areas in (**a**). Conductivity contributions (last term of Eq. (1)) have been subtracted for visual clarity in (**a**, **b**). **c** Shows the constant effective dipole moment of 2Ala $\mu_{eff}$(2Ala) as calculated from the 2Ala relaxation amplitude as a function of 2Ala concentration. Error bars are based on the standard deviation within eight independent measurements.

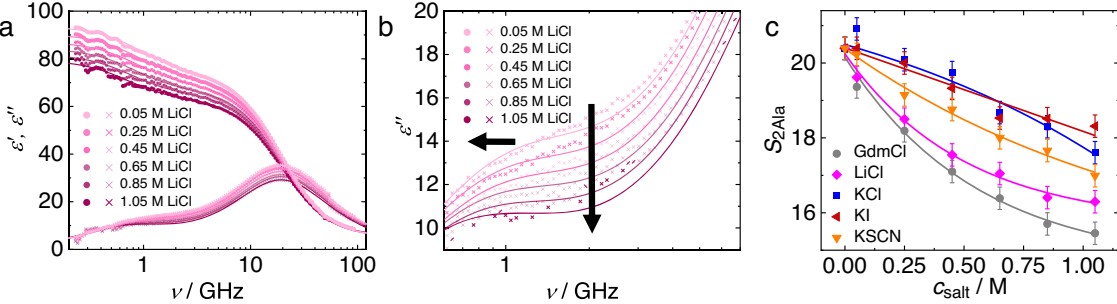

**Fig. 3 Decrease in 2Ala's relaxation amplitude is salt-specific. a** Complex permittivity spectra of aqueous 2Ala solutions (0.25 M) with increasing concentration of LiCl. **b** Zoom into lower-frequency loss spectra, which are dominated by the 2Ala relaxation, with arrows indicating the trend with increasing concentration. Solid circles and crosses show the experimental dielectric permittivity ($\varepsilon'$) and dielectric loss spectra ($\varepsilon''$), respectively. Solid lines show fits of Eq. (1) to the experimental data. Note that the conductivity contribution (last term of Eq. (1)) has been subtracted for visual clarity. **c** 2Ala relaxation strength, $S_{2Ala}$, (Eq. (1)) as a function of $c_{salt}$ for different salts. Error bars correspond to twice the standard deviation from the parameters obtained from eight independent experiments. Lines are guides to the eye.

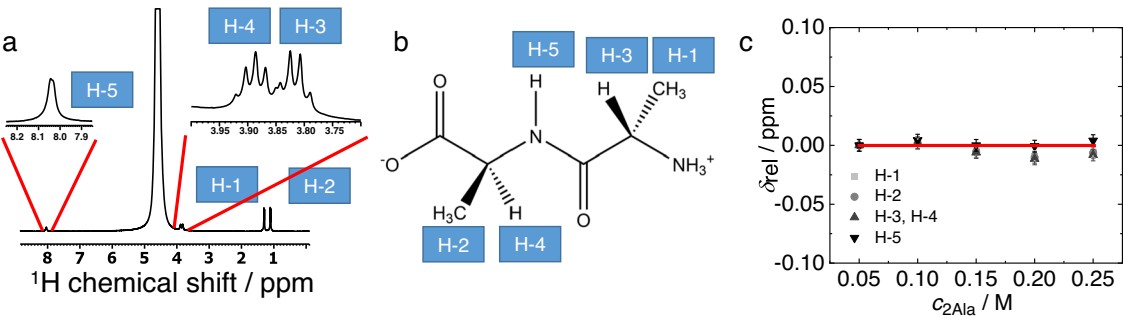

**Fig. 4 ¹H-NMR chemical shifts are independent of 2Ala concentration. a** ¹H-NMR spectrum of 0.25 M 2Ala (in $H_2O$ (DMSO-d5h1 capillary included). Proton signals are assigned to 2Ala's protons as indicated in the structure shown in (**b**). The variation of the chemical shifts $\delta_{rel} = \delta(c_{2Ala}) - \delta(c_{2Ala} = 50\,mM)$ as function of 2Ala concentrations is shown in (**c**). The solid red-line in (**c**) corresponds to $\delta_{rel} = 0$. Error bars show the typical experimental reproducibility of ±0.005 ppm (see the text).

the dielectric loss spectra at frequencies where the 2Ala relaxation dominates (Fig. 3b) shows that also 2Ala's contribution to the experimental spectra decreases with increasing LiCl concentration. Upon the addition of other salts (GdmCl, KCl, KI, KSCN), we find qualitatively similar trends. Yet, the depolarization of the peptide mode can be quantitatively different depending on the nature of the ions[13–16]: Kinetic depolarization of the peptide relaxation requires spatial proximity of the ions to the peptide and, as such, stronger interactions (on average closer proximity) result in a more pronounced reduction of the relaxation strength.

To quantify the ion-specific depolarization of the 2Ala relaxation, we also model the ternary samples with Eq. (1). The parameters obtained from these fits (see Supplementary Fig. 3) show that the 2Ala relaxation slows down with increasing salt concentration (increase in $\tau_{2Ala}$ with increasing salt concentration), which can be related[13,16] to the increasing viscosity with increasing salt concentration[62,63]. For all investigated salts, we find a reduction of the relaxation amplitude $S_{2Ala}$ (Fig. 3c), indicative of the presence of ions in the coordination sphere of 2Ala. We find the effect of the different salts on $S_{2Ala}$ to be ion-specific: KCl and KI reduce the relaxation strength only by ~10% for the highest salt concentration (1.05 M), while for KSCN, we find a reduction by almost 20% (Fig. 3c). As such, the studied denaturing anions moderately reduce the rotational degree of freedom of the model peptide. For the salts with a denaturing cation, LiCl and GdmCl, we find the strongest reduction of $S_{2Ala}$ by up to ~25%. These findings are in broad agreement with what we have concluded for the rotational mobility of N-methylacetamide and triglycine[13–16] in aqueous solution: salts with increasing denaturation tendency

on proteins affect the rotational mobility of model peptides more profoundly, with a more pronounced effect for denaturing cations as compared to denaturing anions. Based on the kinetic depolarization mechanism described above, our findings imply (on average) closer proximity of Hofmeister cations to 2Ala, as compared to Hofmeister anions when approximating 2Ala as a point dipole. Yet, the effect of the finite size of 2Ala, which makes ion-dipole interactions location-specific, is not considered. Therefore, site-specific molecular-level details cannot be pinpointed.

**Chemical environment of 2Ala's protons probed by ¹H-NMR spectroscopy.** To obtain insights into site-specific interactions of ions with 2Ala we probe the chemical environment of 2Ala's protons using ¹H-NMR spectroscopy[31,34]. First, we assign all NMR signals detected in the ¹H-NMR spectra (see Supplementary Figs. 4 and 5) of solutions of 2Ala in $H_2O$ (DMSO-d5-h1 capillary included): Based on ¹³C-NMR spectra (see Supplementary Fig. 6), ¹H-¹³C-HSQC, and ¹H-¹³C-HMBC (see Supplementary Figs. 7 and 8 and Supplementary Discussion 2) we ascribe the NMR peaks at 1.29 ppm and 1.10 ppm to the β protons ($CH_3$ groups) at the N-terminus (H-1) and the C-terminus (H-2), respectively (Fig. 4a, b). The peptide's α-protons at the N-terminal Ala unit (H-3) and at the C-terminal (H-4) alanine group give rise to signals at 3.82 and 3.88 ppm, respectively. We detect the amide proton (H-5) at 8.04 ppm (Fig. 4a, b). As can be seen from the concentration-dependent relative chemical shifts $(\delta_{rel} = \delta(c_{2Ala}) - \delta(c_{2Ala} = 50\,mM)$, Fig. 4c), all these chemical shift values are insensitive to the concentration of 2Ala within the experimental uncertainties

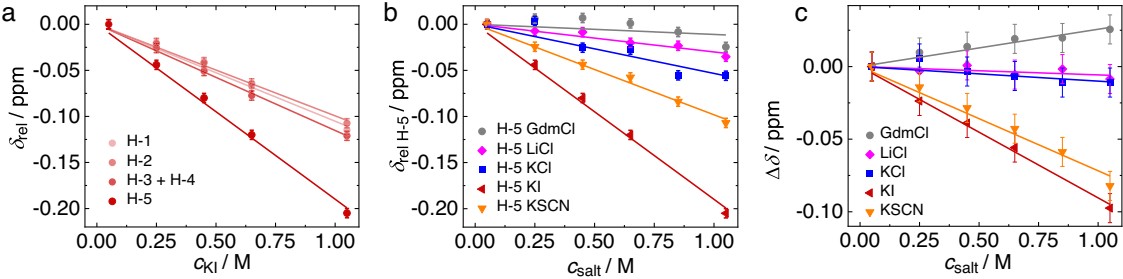

**Fig. 5 Salt-induced changes of the chemical shifts of the amide proton (H-5) differ from aliphatic protons H-1–H-4. a** Concentration-dependent relative chemical shift, $\delta_{rel} = \delta(c_{salt}) - \delta(c_{salt} = 0.05\,M)$ for aqueous solutions of 250 mM 2Ala with increasing concentration of KI. **b** Concentration and salt-dependent relative shift of the NH proton H-5, $\delta_{rel\,H-5} = \delta_{H-5}(c_{salt}) - \delta_{H-5}(c_{salt} = 0.05\,M)$. **c** Differential chemical shift $\triangle\delta = \delta_{rel\,H-5} - \delta_{rel\,H-1}$. Symbols show experimental data and error bars are based on the typical experimental reproducibility. Solid lines show linear fits.

($\pm0.005$ ppm), which is largely due to inaccuracies in referencing the signals to the dimethylsulfoxide included in the capillary (see experimental section). As such, the data in Fig. 4c confirm that 2Ala aggregation can be neglected[64], as already concluded from the dielectric data (Fig. 2c).

Conversely, the chemical shifts vary markedly upon adding salt: as exemplarily shown for KI in Fig. 5a (representative $^1$H-NMR spectra are shown in Supplementary Fig. 9), increasing concentration of salt results in an up-field shift of all five detected proton signals.

In general, such variation of the chemical shift can be due to a change of the solvent properties upon replacing the solvent $H_2O$ for an aqueous salt solution (medium effect)[39,65] or due to specific binding of ions to the peptide[39]. Yet, for weak, specific binding both contributions are challenging to discriminate based on the concentration-dependent shift values as they lead to an approximately linear variation in the chemical shift[66]. In the case of KI, we find the protons H-1–H-4 to exhibit a similar variation of the chemical shifts with increasing salt concentration: The shaded red traces in Fig. 5a showing $\delta_{rel} = \delta(c_{salt}) - \delta(c_{salt} = 0.05\,M)$ for H-1–H-4 nearly overlap. The same applies to the other salts used in the present study (see Supplementary Fig. 10), with an ion-specific slope for the $\delta_{rel}(c_{salt})$ curves. The similar salt-induced shift for H-1–H-4 suggests that the aliphatic protons H-1–H-4 predominantly sense medium effects due to the variation of the solvents electronic properties with increasing $c_{salt}$. Accordingly, the salt specificity of the slopes of $\delta_{rel}$ can be ascribed to the different (electronic) properties of the aqueous salt solutions. In contrast to the protons H-1–H-4, the amide proton H-5 undergoes different changes in the chemical environment and $\delta_{rel}$ of H-5 decreases more steeply for KI, as compared to H-1–H-4. This different sensitivity of H-5 to added KI points towards enrichment or closer proximity of iodide to the amide proton, as compared to the aliphatic protons. We note that we detect virtually the same changes for solutions containing 50 mM 2Ala, suggesting that also in the presence of KI aggregation induced changes to the chemical shift of 2Ala can be neglected (see Supplementary Fig. 11 and Supplementary Discussion 2).

Similar to the findings for KI, we also find for KSCN a marked decrease of the chemical shift of H-5 with increasing concentration (Fig. 5b), while for KCl and LiCl the decrease is only moderate—in line with negligible interaction of these salts with the amide group[24]. For the denaturant GdmCl, we find a weak down-field shift at low-salt concentrations, and a moderate up-field shift at higher salt content. To isolate the salt-specific changes of the amide proton's chemical shift from medium effects[65], we assume the variation of $\delta_{rel}$ of H-1 to be solely due to medium effects. Based on this assumption, the concentration-dependent variation of the difference in the chemical shifts of H-5

and H-1 report on specific effects of the salts on the amide proton: As can be seen in Fig. 5c, this differential chemical shift value $\triangle\delta = \delta_{rel\,H-5} - \delta_{rel\,H-1}$ varies linearly with increasing salt concentration for all studied salts. Due to its differential nature, the shift difference can also account for nonlinear medium effects[21] and reduces the scatter of the data, which originate from inaccuracies in referencing the spectra with the external capillary (see "Methods").

The data in Fig. 5c suggest that—within experimental error—KCl and LiCl do not affect the differential shift, consistent with the notion that the observed variation can be solely ascribed to medium effects[65]. For KI and KSCN, the marked shift of the H-5 signals to lower ppm values with increasing salt concentration is indicative of the interaction of the anions with the amide group: Given the weak hydrogen-bond acceptor strengths of these anions compared to the chloride salts[67], increasing substitution of water molecules by I$^-$ or SCN$^-$ anions is expected to result in enhanced shielding of the NH proton due to the anions themselves and due to weakening of the N–H hydrogen-bond[68]. Interestingly, for GdmCl, the shift difference increases with increasing GdmCl concentration (Fig. 5c). Hence, despite the guanidinium cation being a moderately strong hydrogen-bond donor[69], it apparently can affect the H-bond donating N–H group of 2Ala. This observation may be explained by the interaction of Gdm$^+$ with the amides CO group via bidentate binding to both, the amide and the carboxylate group[16]. Hence, hydrogen bonding of Gdm$^+$ to 2Ala's CO group may also affect the charge distribution at the N–H group such that the N–H proton is de-shielded. Altogether, our NMR results suggest that denaturing anions preferentially accept hydrogen bonds from the amide N–H group. The almost perfect linear variation of the data in Fig. 5c suggests that this binding is weak, and a fourfold excess of salt with respect to 2Ala is not sufficiently high to make the chemical shift values plateau at the value of 2Ala-ion complexes.

**The environment of 2Ala's C–O groups probed by linear infrared and 2D-IR spectroscopy.** To probe salt-induced changes in the vicinity of the carbonyl or carboxylate groups of 2Ala, we use infrared (IR) spectroscopy, which is particularly sensitive to molecular vibrations with a high transition dipole moment[70]. Here, we focus on two vibrational modes, the amide I vibration (CO stretch vibration) at ~1660 cm$^{-1}$ (see ref. [28]) and the anti-symmetric COO$^-$ stretching vibration at ~1590 cm$^{-1}$ (see ref. [71]), which are common vibrational probes for the structure and dynamics of peptides and proteins[27,28,72–77]. These two vibrational modes are well-separated for a 250 mM solution of 2Ala in D$_2$O (Fig. 6a), with the carboxylate mode being somewhat narrower than the amide I vibration. At the highest salt concentration used for the NMR and DRS experiments (~1 M), the salts hardly affect

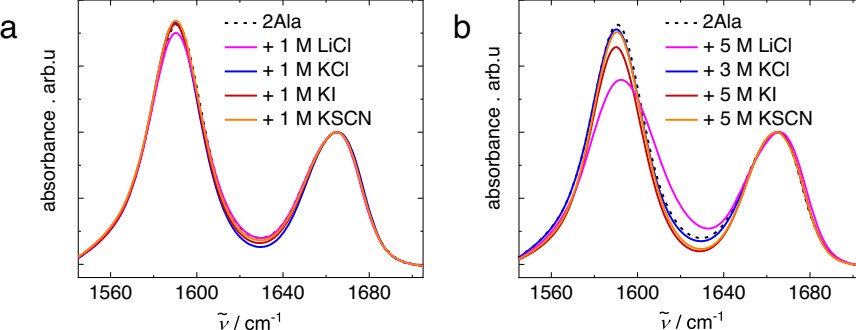

**Fig. 6 Salts hardly affect the IR bands of the amide I and anti-symmetric carboxylate stretching mode.** Infrared absorption spectra in the absence of salt: 250 mM 2Ala in $D_2O$ (dashed black line) and in the presence of (**a**) 1 M and (**b**) 5 M (3 M for KCl) salt. Spectra were corrected for a linear background and normalized to the CO peak at ~1660 $cm^{-1}$.

these vibrational modes: the addition of 1 M LiCl, KCl, KI, or KSCN does not affect the lineshape of both vibrations, and results in only minor changes in the amplitude of the anti-symmetric carboxylate stretching band relative to the amide I mode. We note that the presence of the anti-symmetric CN stretching mode of Gdm+ at ~1610 $cm^{-1}$ (see ref. [78]) prevents us from isolating the carbonyl and carboxylate bands for solutions containing GdmCl (see Supplementary Fig. 12 and Supplementary Discussion 3). Nevertheless, the amide I band appears as a shoulder of the C–N band and coincides with the amide I band in solutions with other salts, suggesting that also GdmCl does not affect the amide I mode of 2Ala significantly. The insensitivity of both vibrational bands in the presence of 1 M salt (Fig. 6a) is consistent with earlier studies[27,28,77], that found salt-induced changes to the vibrational modes only at elevated concentrations. Changes to the IR spectra are hardly detectable for the presently studied salts, also at elevated salt concentrations (5 M for LiCl, KI, and KSCN; 3 M for less soluble KCl, Fig. 6b), except for LiCl, for which the carboxylate band slightly broadens (see Supplementary Fig. S13 and Supplementary Discussion 3). This broadening is also observed for samples containing 50 mM 2Ala, which suggests that it is not related to 2Ala self-aggregation, but can be related to peptide-salt interactions (see Supplementary Fig. 14).

The lineshape of the linear infrared bands can typically be traced to distribution of molecular oscillators in different microenvironments (inhomogeneous broadening). While this distribution is largely unaffected by the salts for the present samples, the dynamics with which these microenvironments interconvert can be altered by salts. These dynamics remain hidden in the linear spectra, but nonlinear infrared spectroscopies can uncover these dynamical aspects. Here, we perform two-dimensional infrared (2D-IR) spectroscopy[70], where a specific vibration is "tagged" via vibrational excitation. For the time-domain experiments used herein, the excitation is generated using two laser pulses, and the excitation frequency is resolved in the time domain. The response of the excited vibrations is probed by a probe pulse, which is detected in the frequency domain. In a typical 2D-IR experiment, the (probe) frequency-dependent response is mapped as a function of the excitation frequency.

Figure 7a shows the 2D-IR spectra of a 0.25 M solution of 2Ala in $D_2O$ right after the excitation (waiting time $T_2 = 0$ fs). For both, the amide I and the carboxylate band, we detect a pair of signals: a negative (shaded blue) signal at the diagonal, at which the excitation frequency equals the detection frequency, due to the population depletion of the vibrational ground state/ stimulated emission from the excited state and a positive signal (shaded red, induced absorption) due to the anharmonically red-shifted excited state absorption.

With increasing waiting time, the intensities of these signals decrease due to relaxation to the vibrational ground state. This decrease is more pronounced for the carboxylate band than for the amide I mode, demonstrating faster vibrational relaxation of the $COO^-$ mode (see, e.g., Fig. 7b). To quantify vibrational relaxation dynamics, we extract the peak volumes for both modes as a function of delay times (see Supplementary Fig. 15 and Supplementary Discussion 4). These volumes decay with a characteristic vibrational relaxation time $\tau_{VER}$ of ~0.4 ps for the anti-symmetric $COO^-$ stretching band and ~ 0.7 ps for the amide I mode. The latter is about twofold faster than what has been reported for 2Ala or 3Ala at low pH, at which the carboxylate group is protonated[79]. The faster relaxation observed here is presumably related to the altered vibrational and electronic structure of 2Ala upon deprotonation[80], but also coupling between the carboxylate and amide I mode can constitute a relaxation channel[81]. In fact, we detect low-intensity cross-peaks for both modes (see Fig. 7b), peaking at ~200 fs waiting time, consistent with vibrational energy transfer (see Supplementary Fig. 16). As such, coupling with the $COO^-$ group appears to provide a relaxation path from the excited state of the amide I mode. Yet, within experimental accuracy, we find no indications for ion-specific effects on the vibrational relaxation dynamics (see Supplementary Fig. 17) or the cross-peaks dynamics/intensities (see Supplementary Fig. 16), which indicates that the addition of ions negligibly influences the coupling of the vibrations to the environment (Supplementary Discussion 4).

The spectral shape of the detected signals, however, depends on both the waiting time and the added salt. In general, the frequency response of the sample is correlated to the excitation frequency for an inhomogeneously broadened band, which results in an elongation of the 2D-IR signals along the diagonal[70]. For 2Ala in the absence of salt, the signals due to the CO stretch vibration (at 1660 $cm^{-1}$) are markedly elongated along the diagonal, indicative of inhomogeneous broadening, while the anti-symmetric $COO^-$ stretching mode (at 1590 $cm^{-1}$) appears more parallel to the pump axis. With increasing waiting time, as shown exemplarily for 2Ala at 500 fs in Fig. 7b (for other concentrations and waiting times, see Supplementary Figs. 18–29), the correlation between excitation and probing frequencies is reduced due to spectral diffusion, as evident from the increased vertical elongation of the detected signals[70].

The addition of salt can alter these correlations and their decay dynamics: In the presence of 5 M LiCl, the $COO^-$ signal is markedly elongated along the diagonal (Fig. 7c), and the correlations partly persist up to a waiting time of 500 fs (Fig. 7d). In contrast to the observations for LiCl, the 2D-IR spectra of 2Ala in the presence of 3 M KCl (Fig. 7e, f), 5 M KI (Fig. 7g, h), and

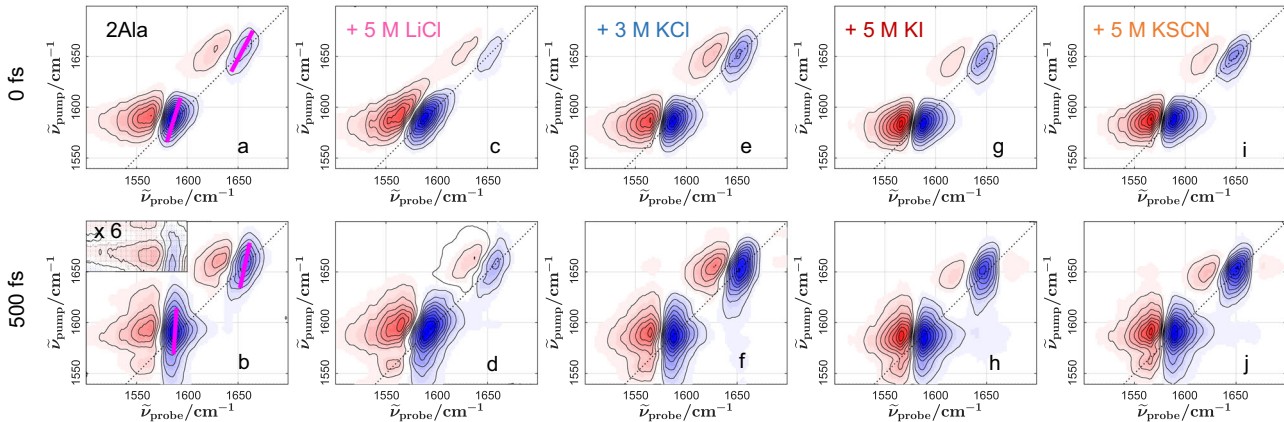

**Fig. 7 2D-IR spectra of 2Ala + salt to reveal the ion-specific vibrational structure and spectral dynamics.** 2D-IR spectra of 0.25 M 2Ala in $D_2O$ (**a, b**) in the absence of salt, with 5 M LiCl (**c, d**), with 3 M KCl (**e, f**), with 5 M KI (**g, h**) and 5 M KSCN (**i, j**). Top panels show spectra at 0 fs waiting time. With increasing waiting time, vibrational relaxation results in a decay of the signals, leading to a variation of the relative intensities of both diagonal peaks at a waiting time of 500 fs (bottom panels). The dotted black lines indicate the diagonal line. The inset in (**b**) shows a zoom into the region where the cross-peak is detected. Pink lines in (**a, b**) represent the center line slope used to quantify frequency–frequency correlations.

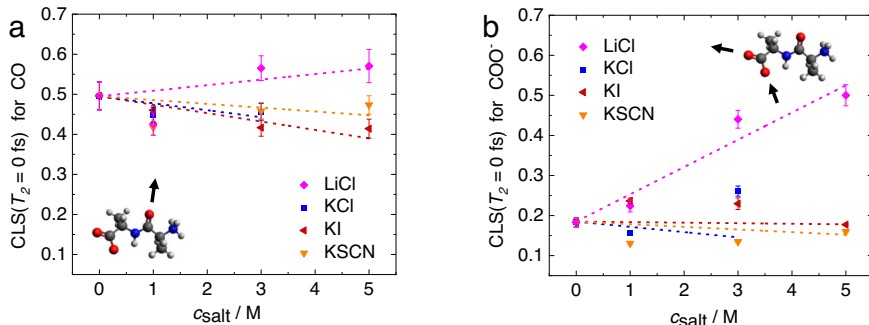

**Fig. 8 Spectral heterogeneity is salt-specific for the $COO^-$ mode but not for the amide CO.** Center line slope at 0 fs waiting time for **a** the amide CO and **b** the anti-symmetric $COO^-$ stretching mode. Symbols show experimental data, dashed lines are a visual aid. Error bars correspond to the standard deviation of the CLS of the bleaching signal determined from five different scans.

5 M KSCN (Fig. 7i, j) qualitatively resemble the spectra for 2Ala in the absence of salt (Fig. 7a, b).

To quantify the heterogeneity of the detected signals and the decay dynamics of the correlations (spectral diffusion), we determine the waiting-time-dependent center line slopes (CLS, see Supplementary Discussion 4) and fit an exponential decay $(CLS(T_2) = CLS_0 \exp(-k_j T_2))$ to the data (see Supplementary Figs. 30–33)[82]. The instantaneous ($T_2 = 0$ fs) center line slope of the amide I mode in the absence of salt is determined to ~0.5 (Fig. 8a), which agrees well with the value reported for protonated 2Ala[79]. From single exponential fits to the waiting-time-dependent CLS, we also find the decay time of ~1.1 ps (see Supplementary Fig. 31b) for the amide I mode to be comparable to protonated 2Ala[79,83]. This similarity suggests that the exchange of microenvironments of the amide CO, which stems from the hydrogen-bond dynamics of the hydrating water molecules, is hardly affected by protonation at the C-terminus.

We find that the presence of salts hardly affects the instantaneous heterogeneity of the CO mode (Fig. 8a): addition of all potassium salts results in a (<10%) decrease of the $CLS(T_2 = 0)$, which may be explained by salts altering the hydration structure of the amide group due to the altered osmolality. Upon addition of LiCl, we find a minor increase in $CLS(T_2 = 0)$. Overall, the salt-induced variations in the spectral heterogeneity are very modest, and none of the presently studied salts significantly alters the associated spectral diffusion dynamics (see Supplementary Fig. 31b). As such, our results imply that the

environment of the CO groups in the aqueous salt solutions is comparable to its environment in aqueous 2Ala solutions. This notion is broadly consistent with the earlier finding that cations interact only weakly with amide groups[38].

In turn, the CLS at 0 fs for the anti-symmetric carboxylate stretching vibration is markedly altered in a salt-specific manner (Fig. 8b): While we find the detected instantaneous heterogeneity is nearly unchanged even at $c_{salt} = 5$ M KI, KSCN, and 3 M KCl, the $CLS(T_2 = 0)$ increases by a factor of ~2.5 upon addition of 5 M LiCl. As such, the modest changes to the infrared bands (Fig. 6) originate from a marked change in the environment of the $COO^-$ group in the presence of LiCl. These profound changes upon addition of LiCl presumably stem from a slow-down of the fast (within our experimental time resolution) spectral diffusion dynamics, which also affect the magnitude of $CLS(T_2 = 0)$. Indeed, within experimental error, KCl, KSCN, and KCl do not affect the detected CLS decay dynamics (see Supplementary Fig. 31a), but the CLS (~300 fs decay time in the absence of salt) decays markedly slower (~1.3 ps decay time) in the presence of 5 M LiCl (see Supplementary Figs. 31 and 34). Together, these observations suggest that LiCl markedly alters the microenvironments and their dynamics for the $COO^-$ group: $Li^+$ cations interact with the negatively charged carboxylate group. The enhanced tendency of $Li^+$ to interact with 2Ala's carboxylate group is in line with the general tendency of alkali cations to bind to carboxylates[84]. In this context, it is interesting to note that for the interaction of $Li^+$ with carboxylates direct ion contacts have been suggested to play only a

minor role, rather solvent-separated contacts prevail[84]. Our observation that LiCl can efficiently alter the vibrational structure of 2Ala's carboxylate group may thus only partially stem from direct contacts. Yet, also long-lived contacts with hydrated ions could alter 2Ala's vibrational dynamics, as water in ionic hydration shells is less dynamic than in bulk water.

**Conclusion.** In summary, we studied the effect of salts on the environment and dynamics of alanyl-alanine in water. Our DRS results suggest that salts with increasing protein denaturation tendency can increasingly restrict the diffusive rotational motion of the dipolar 2Ala zwitterion. We find denaturing cations, like $Gdm^+$ and $Li^+$ to be most efficient in reducing the rotational relaxation of 2Ala, indicative of closer proximity of these cations to 2Ala in solution. Denaturing anions, like $I^-$ and $SCN^-$, also affect 2Ala's diffusive rotation, however, to a lesser extent. $^1$H-NMR chemical shift experiments evidence that salts induce a change in the chemical environment of all protons of 2Ala. This change can, however, largely be explained by a bare medium effect, except for 2Ala's amide proton: the chemical shift of the amide proton is very sensitive to the presence of KI and KSCN. This increased sensitivity suggests that the denaturing $I^-$ and $SCN^-$ anions interact with the amide N–H group. Conversely, for GdmCl, the shift variation is reversed, which— in contrast to $Li^+$ and $K^+$—points toward the binding of $Gdm^+$ to the amide CO. Our 2D-IR results show that both the infrared absorption spectra at amide I and anti-symmetric carboxylate stretching frequencies and the spectral dynamics of the amide I band are rather insensitive to the presence of salts, and the salt-induced changes to the vibrational structure and dynamics are similar for all salts, irrespective of their denaturation tendency. Conversely, we find the salt-induced changes to the vibrational structure and dynamics of the $COO^-$ group to be salt-specific: we find the strongest effect for LiCl, for which the spectral heterogeneity and its associated spectral diffusion dynamics are markedly slowed down. Although previous reports have indicated that ions like $Li^+$ can bind to the amide group[18,31], and that for an isolated amide group in N-methylacetamide cations can perturb the amide more efficiently than anions[13], our results show that in the presence of an amide group and a C-terminal $COO^-$ group, the carboxylate group is the prevailing interaction site for $Li^+$.

Based on our findings for 2Ala, our results thus provide a unifying view of ion model-peptide interactions, for which previous experiments based on different experimental techniques have led to partially inconsistent conclusions, in particular related to the interaction strengths of anions compared to cations. We find that anions preferably interact with the N–H group of 2Ala. This interaction site can explain why—despite their high denaturation efficiency towards proteins[18]—anions are less efficient in reducing the rotational mobility of small model peptides as detected with DRS. The N–H group is located close to the geometric center (i.e., rotational axis) of the 2Ala dipole, but more distant from the ionic sites—the poles of the dipole. At this location, ion-dipole interactions are weaker as compared to ions in the proximity of the charged sites. The opposite argument holds for cations: interaction of the cation with the charged carboxylate group is consistent with stronger ion-dipole interactions. Our infrared experiments confirm that cation–carboxylate interactions prevail over amide CO—cation interactions. This interaction with the carboxylate presumably has only limited effect on the backbone of a real protein, but may be relevant to the interactions of charged residues that stabilize proteins[85]. Also for the self-assembly of short peptides charged sites can play a key role[86], and our results suggest that LiCl can efficiently distort such interactions. More general, the herein-found preference of anions to the amide group and the

preference for cations to the carboxylate groups may help rationalize why denaturing anions are often more efficient denaturants than cations: amide groups are typically much more abundant in a protein than carboxylate groups.

## Methods

**Sample preparation.** Guanidinium chloride (GdmCl), LiCl, KCl, KSCN, KI, and L-alanyl-L-alanine (2Ala) were used without further purification. Solutions for DRS and NMR experiments were prepared using $H_2O$ with a specific resistivity >18 $M\Omega cm^{-1}$ from a Millipore MILLI-Q purification unit or using $D_2O$ for infrared experiments. All samples were prepared by weighing the appropriate amount of salt and peptide into a 1 mL volumetric flask using an analytical balance. The peptide concentration was kept constant at 250 mM for all samples, while the salt concentration was varied from 0.05 M to 1.05 M for DRS and NMR experiments. For infrared experiments, we used salts concentrations of 1.05, 3, and 5 M.

For DRS experiments, 1 mL of each sample was brought in contact with the coaxial probes. For NMR experiments, 0.5 mL of the samples were placed in an NMR tube together with a capillary filled with DMSO-d6 for referencing. For all infrared experiments, solutions were contained between two $CaF_2$ windows (2 mm thickness, 2.54 cm diameter) separated by a 25 μm Teflon spacer. The infrared absorption spectra were measured right before collecting the 2D-IR data.

**DRS measurements.** Dielectric permittivity spectra were measured using an Anritsu MS4647A vector network analyzer in the frequency range 0.25–125 GHz. To cover the frequency range at 0.25–54 GHz, a frequency domain reflectometer based on 1.85 mm coaxial connectors and an open-ended probe head was used[87,88]. For the frequency range 54–125 GHz, an external frequency converter (Anritsu 3744 A mmW module) connected to a 1 mm open-ended probe was used. To calibrate the instrument for directivity, source match, and frequency response errors, three different references were measured on the same day as the samples: open (air), short (conductive silver paint), and load (pure water). The sample temperature was controlled to 25 ± 1 °C using a Julabo F12-ED thermostat. All measurements at both probe heads were performed eight times per sample.

**NMR measurements.** All NMR spectra were recorded using a Bruker DRX 400 MHz Spectrometer equipped with a commercial Bruker 5-mm 2-channel inverse probe head with z-gradients. $^1$H and $^{13}$C spectra were obtained at room temperature with the standard pulse sequences and parameters. The number of scans ranged from 8 to 128, depending on the concentration of the samples. All spectra were referenced using the residual DMSO-$d_5h_1$ signal at 2.5 ppm[89]. The absolute chemical shifts of 2Ala were extracted using Topspin 3.6.1.

Gradient-enhanced two-dimensional (2D) $^1$H–$^{13}$C HSQC experiments were performed using the hsqcetgp pulse program with the following acquisition parameters: F2 and F1 spectral widths of 20.61 ppm and 179.99 ppm and a F2 and F1 resolution of 8.06 Hz/pt and 70.78 Hz/pt, respectively. In total, 256 FIDs were recorded, each consisting of 32 scans and 2048 data points. The relaxation delay was set to 2 s. Gradient-enhanced 2D $^1$H–$^{13}$C HMBC experiments were based on the hmbcgpndqf pulse program with the following acquisition parameters: F2 and F1 spectral widths of 20.61 ppm and 239.99 ppm and F2 and F1 resolution of 4.03 Hz/pt and 94.38 Hz/pt, respectively. In total, 256 FIDs were recorded, each consisting of 16 scans and 2048 data points. The relaxation delay was set to 2 s.

**FT IR measurements.** Linear infrared absorption (IR) spectra were measured using a Bruker Vertex 70 spectrometer in transmission geometry. The spectra were recorded with a resolution of 4 $cm^{-1}$ at frequencies ranging from 400 to 4000 $cm^{-1}$. The sample compartment was purged with nitrogen during the measurement.

**2D-IR measurements.** The two-dimensional infrared setup is based on 800 nm laser pulses (7 W, 35 fs, 1 kHz) from a regenerative amplifier laser system (Coherent, Astrella). In all, 2.7 W of the 800 nm pulses were used to pump an optical parametric amplifier Topas Prime (Coherent) to generate signal and idler pulses. Signal and idler pulses were used to generate infrared pules at ~6000 nm (18 μJ, 400 $cm^{-1}$ FWHM) using non-collinear difference frequency generation (NDFG) Topas (Coherent). The IR beams are guided into a commercial 2D infrared spectrometer 2D Quick IR (Phasetech, Inc.).

In the spectrometer, a weak reflection[90,91] from a wedged ZnSe window is used as probe beam. The transmitted IR light is guided to a pulse shaper, where it is diffracted from a grating (150 l/mm), collimated using a parabolic mirror, and guided to a Germanium-based acousto-optic modulator (AOM). The IR light is diffracted at the AOM and focused onto a second grating (150 l/mm). The shaped beam is reflected from a retroreflector on a translational stage to control the waiting time ($T_2$) between the pump and the probe beams. After setting the polarization of the pump beam to 45° relative to the probe beam polarization, the pump and the probe beams are focused into the sample using an off-axis parabolic mirror. After the sample, the probe beam is re-collimated and split into polarization components, parallel and perpendicular relative to the pump beam with a polarizer. Both probe components are focused into an imaging spectrograph

(SP2156 spectrograph, Princeton Instruments, 30 l/mm grating) and detected using a $128 \times 128$ pixel mercury cadmium telluride (MCT) array detector.

Frequency resolution on the pump axis is obtained in time domain, using the pulse shaper to generate two pump pulses delayed by 0 to 2555 fs at increments of 35 fs. To reduce data acquisition time we used a rotating frame at $1400\ cm^{-1}$. Pump pulses were corrected for group-velocity dispersion, and the dispersion parameters were optimized using the transient signal generated from multi-photon absorption in a 0.5 mm thick Ge plate. Before transforming the thus determined free induction decays into the frequency domain, the time-domain data were zero-padded to 128 data points and filtered with a Hamming window.

## Data availability

The datasets generated during the current study are available from the corresponding author upon reasonable request.

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

## Acknowledgements

We are grateful to Mischa Bonn and Yuki Nagata for insightful discussions. This project has received funding from the European Research Council (ERC) under the European Union's Horizon 2020 research and innovation program (grant agreement no. 714691).

## Author contributions

C.K. and J.H. designed the experiments and the research project. C.K. prepared the samples and performed the DRS measurements. C.H. and C.K. performed the NMR measurements. L.G. and C.K. performed the IR measurements. All authors discussed the results. C.K. and J.H. wrote the manuscript.

## Funding

## Competing interests

The authors declare no competing interests.
