## [Peer Review File · Communications Chemistry]

Ion-specific binding of cations to the carboxylate and of anions to the amide of alanylalanineReviewers' comments:

Reviewer #1 (Remarks to the Author):

The article presents a study of the structure and dynamics of L-Alanyl-L-alanine 13 (2Ala) in aqueous ionic solutions by dielectric relaxation (DA), NMR, and IR (including 2DIR) spectroscopic methods. The authors argue that 2Ala is a good model system for understanding the Hofmeister effects on proteins. However, this reviewer finds that 2Ala is not a sound model for such studies. It is a zwitterion and contains only an amide group between the charged species at both ends. The positive and negative charges are closely located, and the structure and dynamics of 2Ala are likely to be strongly influenced by the Coulombic force between the two charged species. Therefore, the results obtained in this study do not seem very closely related to proteins composed of many amino acids.

There are also multiple technical issues that need to be addressed.

Technical issues that need to be addressed:

1. The 250 mM concentration is too high to avoid aggregates. It is recommended to check concentration-dependent IR spectra to ensure that there are no aggregates.
2. In line 306 – 307 : The sentence makes us think that 2DIR spectroscopy is a 2nd-order nonlinear experiment. A single infrared laser pulse does not generate population states. Clarification is needed.
3. The quality of 2D IR spectra shown in Figures 7, S12, and S13 is not sufficient to accurately assess the peak shape in the spectra, eventually leading to the inability to extract the structure and dynamics of 2Ala.
4. The CLS assessment shown in Figure S16, in particular b), does not match with the fits. Each data set was fitted with a single exponential function. However, the experimental data sets show a sigmoidal pattern. Could there be artifacts or some other unknown dynamics? In Figure S12 h), there are two blue peaks on the lower right corner of the spectrum. They suggest the diagonal peak at $\sim 1585 \text{ cm}^{-1}$ consists of two peaks. In this case, the CLS analysis is supposed to be contains a serious error. It is recommended to present the 2D IR spectra used in evaluating the CLS. About a quarter (if not all) of the obtained spectra, will give the readers a clear understanding of this article.

Reviewer #2 (Remarks to the Author):

Hunger and coworkers investigated the influence of salts on a peptide consisting of two alanine residues, 2Ala. Measurements were made in water with DR, NMR, IR, and 2D-IR. The author wanted to investigate conflicting conclusions as to whether cations or anions interact more strongly with peptides with the stated purpose of helping to "rationalize Hofmeister effects on proteins." Generally, I like the authors multi-spectroscopy approach and it is nice to see side-by-side data using DR, NMR, and vibrational spectroscopy on the same small molecule. I also agree with much of the local spectroscopic interpretations that the authors conclude. Despite this enthusiasm for the approach and data analysis, the authors have misinterpreted the meaning of some of their data. This will need to be corrected before this paper could be considered ready for publication. My detailed comments are as follows:

- (1) The most important problem is the authors misconception that 2Ala contains "all fundamental protein motifs: C-terminus, amide backbone, and N-terminus." In fact, 2Ala is too small to stand in for the interactions of anions with a protein backbone. Most proteins are significantly longer, and the difference is crucial to understanding Hofmeister chemistry. The water structure around dipeptides contain fewer broken hydrogen bonds compared to larger polymers that have the same functional groups (ref. 59 as well as Thosar, Nat. Chem. 14, 2022, 8-10). Weakly hydrated anions like SCN⁻ bind more strongly at positions where the hydrogen bonding is broken.

(2) The authors found that the 2Ala amide NH is very sensitive to the presence of KI and KSCN. They therefore concluded that the abundance of these sites on proteins would be the key reason that denaturing anions are more efficient than cations. For longer chain polymers, however, previous work has shown that when the NH group on an amide is alkylated, that denaturing anions still bind to roughly the same extent as when the NH group is present (Rembert, Langmuir, 31, 2015, 3459-3464). One possibility is that the NH group might interact even more strongly with the I⁻ or SCN⁻ if the chain were simply longer. Another is that broken hydrogen bonding might lead to better interactions with the two methyl groups or the alpha carbon on 2Ala.

(3) The data in Figures 2 and 4 are supposed to assure the reader that 250 mM 2Ala is a sufficiently low concentration that interactions between dipeptides can be ruled out in the subsequent work with salts in Figures 3, 5, and the infrared spectra. However, just because these peptides do not interact with each other in the absence of salt, does not mean that molar quantities of salt are not causing the dipeptide to associate with itself. The best way to test this would be to spot check the DR and NMR data with a lower concentration of 2Ala. This should be done down as close to the detection limit as possible.

In summary, this paper is potentially interesting because using 4 different spectroscopies to study the same small molecule can potentially help the Hofmeister community gain a better appreciation of the differences in information that each technique can provide about ion interactions with dipeptides. However, the authors stated that they wanted to resolve the "orthogonal conclusions" that "have been obtained with either cations or anions interacting more strongly with model peptides." It would be best for the authors to simply acknowledge that NMA (ref. 14) and 2Ala are too short to provide proper information on the interactions of denaturing anions with the polypeptide backbone of full length proteins.

Reviewer #3 (Remarks to the Author):

In this manuscript, the authors utilize a trio of spectroscopies to answer the age old question of the salt effects on the peptide backbone. It has been brought into question several times during discussion on folding and dynamics. The authors use 2D IR spectroscopy to directly detect any changes on the fast dynamics of the changing electric fields around both the amide I CO and the C-terminal COO to characterize the weak interactions with different salt concentration. The first two spectroscopies highlight both those denaturing cations and anions have significant effects to different areas of the peptide. Using 2D IR, the authors directly observe and measure that the salt has minimal effect on the vibrational relaxation and the correlation decay. This hints that the fast dynamics remain virtually unaffected. However, one exception is Li⁺ ions. These ions interaction strongly with the terminal COO in a concentration dependent way. These observations provide insights into microenvironments and preferential solvation responsible for weak interaction. The overall insights of this paper are both interesting and significant to contribute to field of peptide studies and thus the paper should be accepted upon addressing the following concerns and revisions:

- a) Although spectral shifts are not always detectable, the reviewer wonders how different the bandwidth is in the linear IR measurements since the bandwidth has contributions of CLS and vibrational relaxation.
- b) In line 326-327, the authors mention energy transfer taking place, yet the authors do not address any influence of the Li⁺ ions on the transfer rates. In other words, does the relaxation channel opened by the coupling increase with Li⁺ concentration? If not, why?
- c) The dynamics of folding and activity would most likely be influenced by amide I interactions. The authors should discuss what influence if any that interaction with the termini would cause. How does

this apply to the bigger picture?

d) The author uncover from the 2D IR measurement that Li^+ is the ion having the greatest interaction with the termini. What is special about the Li^+ ions that make the interaction possible?

Reviewer 1

The article presents a study of the structure and dynamics of L-Alanyl-L-alanine 13 (2Ala) in aqueous ionic solutions by dielectric relaxation (DA), NMR, and IR (including 2DIR) spectroscopic methods. The authors argue that 2Ala is a good model system for understanding the Hofmeister effects on proteins. However, this reviewer finds that 2Ala is not a sound model for such studies. It is a zwitterion and contains only an amide group between the charged species at both ends. The positive and negative charges are closely located, and the structure and dynamics of 2Ala are likely to be strongly influenced by the Coulombic force between the two charged species. Therefore, the results obtained in this study do not seem very closely related to proteins composed of many amino acids.

Reply:

We fully agree with the reviewer that a real protein is much more complex than the 2Ala dipeptide of the present study. For a protein, containing many amino acids, the secondary structure, the structural flexibility, and the concomitant accessibility of protein sites to both, ions and water, will affect specific ion-effects. Yet, the conformational flexibility of a protein or macromolecular model system imposes challenges on isolating the interaction of the binding sites: Upon addition of different ions to the samples, a change in an experimental observable (e.g. the NMR chemical shift of a proton) may stem from (i) direct interaction (e.g. proximity of an ion to the proton) or from (ii) an indirect effect of the salt on the conformation (e.g. salt-induced change of the macromolecule's conformation). As such, without additional information on the conformation, conformationally flexible model systems complicate isolating interactions with ions, which is the scope of the present study.

To clarify the experimental challenges with conformationally flexible protein models, we discuss these in a separate paragraph in the introduction of the revised manuscript, which reads:

Above conclusions have been drawn using a variety of model compounds including simple molecules like N-methyl-acetamide,^{14,36-38} short oligo-peptides,^{8,39,40} and polymers.^{21,24,38,41,42} Comparison of the results using small molecules to those obtained with macromolecules^{21,38} has suggested that the length of the macromolecule largely impacts the interaction with ions. This different behavior has been explained by the different hydration structure of small molecules and macromolecules.²¹ Conversely, using large, conformationally flexible molecules makes it more challenging to isolate ion interactions: an ion-induced change of an experimental observable may stem from direct interaction of an ion with the model system or from an indirect, ion-induced change in the conformation.⁸ For instance, hydration of a salt may lead to dehydration and conformational variation of a macromolecule.⁴³

We further emphasize that a single amide bond cannot fully account for all details of the amide backbone of a protein:

Although 2Ala contains only one amide bond, and thus cannot fully represent the amide backbone of proteins, 2Ala has a rather rigid conformation,⁵¹ which limits changes in the spectroscopic observables due to conformational changes and thus allows isolating ionic interactions spectroscopically.

There are also multiple technical issues that need to be addressed. Technical issues that need to be addressed:

1. The 250 mM concentration is too high to avoid aggregates. It is recommended to check concentration-dependent IR spectra to ensure that there are no aggregates.

Reply:

We thank the reviewer for this comment. To ensure that aggregation of 2Ala molecules does not affect the infrared spectroscopy results, we have recorded absorption spectra for samples at a 5 times lower concentration of 2Ala (50 mM). As also pointed out by reviewer 2 below, salt induced aggregation has to be considered. Therefore, we have recorded spectra for 50 mM 2Ala with varying concentration of LiCl. As can be seen in Figure R1 below, the normalized spectra of solutions containing 50 mM 2Ala coincide with the spectra of samples at 250 mM 2Ala. From this comparison we conclude that aggregation of 2Ala does not affect the carboxylate and amide I vibrations.

We have included this discussion in the main text of the revised manuscript and have added this figure to the revised SI (Figure S14).

Figure R1: Infrared absorption spectra at amide I and asymmetric carboxylate stretching frequencies for solutions of a) 0.25 M 2Ala and b) 0.05 M 2Ala in D_2O with increasing concentration of LiCl. Spectra were normalized to the amide I mode at $\sim 1650\text{ cm}^{-1}$ after subtraction of a linear solvent background. The resemblance of the spectra in panels a) and b) demonstrates that a potential salt induced aggregation of 2Ala does not affect the line shape.

2. In line 306 – 307 : The sentence makes us think that 2DIR spectroscopy is a 2nd-order nonlinear experiment. A single infrared laser pulse does not generate population states. Clarification is needed.

Reply:

We partially disagree with this statement of the reviewer that “a single infrared laser pulse does not generate population states.” A single laser pulse can generate population states, as it is routinely done in infrared pump-probe experiments or in frequency-domain 2D-IR spectroscopy. Yet, generation of a

population state requires two interactions with an electric field. To avoid any potential confusion, we have revised the statement as follows:

Here, we perform two-dimensional infrared (2D IR) spectroscopy,^{71,80} where a specific vibration is 'tagged' via vibrational excitation. For the time-domain experiments used herein, the excitation is generated using two laser pulses and the excitation frequency is resolved in the time-domain. The response of the excited vibrations is probed by a probe pulse, which is detected in the frequency domain. In a typical 2D-IR experiment, the (probe) frequency-dependent response is mapped as a function of the excitation frequency.

3. The quality of 2D IR spectra shown in Figures 7, S12, and S13 is not sufficient to accurately assess the peak shape in the spectra, eventually leading to the inability to extract the structure and dynamics of 2Ala.

Reply:

We appreciate the reviewer's comment, albeit it is unclear what aspects of the quality of the spectra the reviewer refers to. We presume the reviewer refers to the line shape of the asymmetric COO⁻ band at a waiting time of 500 fs. Indeed, at this waiting time the transient signals have already markedly decayed (due to the short vibrational lifetime), which makes the instrument noise to contribute at later times. This instrument noise can indeed distort the line shape at long waiting times. Yet, this distortion negligibly affects our main conclusions for the following reasons:

- (i) All main conclusions on the spectral heterogeneity in the main manuscript can be drawn based on the 2DIR spectra at 0 fs waiting time (see e.g. CLS(0 fs) in Figure 8), at which the signal to noise ratio is sufficiently high so that the 2D-IR line shape can be accurately extracted.
- (ii) The reviewer is correct that the distortion at later times can, in principle, bias the extracted dynamics, e.g. the center line slope dynamics, which we present in the supporting material. The experimental noise at later delay times, however, predominantly stems from pulse-to-pulse energy fluctuations of the probe pulse. Though this can be partly corrected for, it results in noise on the (time-domain) free induction decay data, which is nearly constant at all probing frequencies. As such, probe-pulse fluctuations lead to distortions at late waiting times, parallel to the probe axis. For determination of the center line slopes, we determine the minimum of the transient signal at a constant pump frequency. Such, the noise due to the probe pulse fluctuations does not affect the CLS and the CLS dynamics can be reliably extracted also at longer waiting times. This is evidenced by the smooth monotonic decay of the CLS for the COO⁻ band as a function of waiting time.

To better explain these different aspects of the contributing noise, we have added this notion to the revised SI, which reads as follows:

To determine the center line slopes (CLS), we take slices parallel to the probe axis at frequency ranges 1575 – 1610 cm⁻¹ for the COO⁻ mode and at 1640 – 1670 cm⁻¹ for the CO mode. We determine the minima of the bleaching signal at a given pump frequency. Such, instrument noise due to pulse-to-pulse energy fluctuations of the probe pulse, which can lead to a distortion of the line shapes parallel to the

probe axis, do not affect the position of the determined minima, as the probe pulse energy is constant at a given pump-frequency.

4. The CLS assessment shown in Figure S16, in particular b), does not match with the fits. Each data set was fitted with a single exponential function. However, the experimental data sets show a sigmoidal pattern. Could there be artifacts or some other unknown dynamics?

Reply:

The reviewer raises an excellent point and we agree that a simple single exponential fit is insufficient to capture all features of the CLS dynamics. The CLS dynamics indeed seem to contain an oscillatory feature, in addition to the decay. The focus of the present study is the effect of ions on the vibrational structure and dynamics. We did not detect any clear ion-specific changes to the CLS dynamics for the amide I mode, in contrast to the marked, ion-specific changes of the COO⁻ mode. Therefore, we used for the sake of simplicity a single exponential decay to describe the CLS dynamics of the amide I mode.

We agree that the exponential fit is a simplification. The oscillatory behavior of the CLS decay points towards coupling of a lower frequency mode to the amide I mode. From the CLS data we estimate an oscillation period of ~600 fs (~2 THz). To further test the presence of such coupled vibration affecting the CLS, we have performed new, comparative experiments and determined the CLS for 2Ala at pD 7, 2Ala at pD 1 (where the carboxylate group is protonated), and for N-methylacetamide dissolved in D₂O. We show these additional data in Figure R2. As can be seen from this figure, the oscillatory dynamics are absent for NMA in D₂O, consistent with earlier studies (*J. Phys. Chem. B* 2005, 109, 21, 11016–11026). Also for 2Ala at pD 1, where the carboxylate group is protonated (-COOH), the damped oscillation is hardly detectable. The new experiments confirm the presence of oscillatory dynamics for 2Ala at pD 7. As such, the coupled low frequency mode (or the coupling strength) appears to be intimately related to the presence of the carboxylate group.

The presence of such oscillatory dynamics is interesting in itself. We have attempted to quantitatively describe these dynamics, however, the current time-window – mostly limited by the vibrational relaxation time – does not allow for an in-depth quantification. As such, we keep the simplified description with an exponential decay. Yet, we have included the data shown in Figure R2 below (Figure S31) in the revised SI and added the discussion of the oscillatory dynamics, which reads:

We note that a single exponential decay does not model all features of the CLS dynamics and the data in figure S31 b suggest the presence of oscillatory dynamics, in addition to the decay dynamics. Such oscillatory dynamics indicate that the amide I mode is coupled to a lower frequency vibrational mode, similar to what has been found for water.¹⁰ Interestingly, this coupling is absent for N-methylacetamide, in line with earlier work,¹¹ and is also hardly detectable for 2Ala at pD 1, at which the carboxylate group is protonated (figure S32). This comparison suggests that the coupling to the lower frequency vibration is intimately connected to the presence of the carboxylate group. Yet, due to the limited time window (given by the vibrational energy relaxation time) and the insensitivity of the CLS dynamics of the amide I mode to the addition of salts, we refrain from a more quantitative analysis of this oscillatory feature.

Figure R2: Center line slopes of the amide I mode of 2Ala at pD 7, 2Ala at pD 1, and N-methylacetamide (NMA) dissolved in D_2O as a function of waiting time.

In Figure S12 h), there are two blue peaks on the lower right corner of the spectrum. They suggest the diagonal peak at ~ 1585 cm^{-1} consists of two peaks. In this case, the CLS analysis is supposed to be contains a serious error. It is recommended to present the 2D IR spectra used in evaluating the CLS. About a quarter (if not all) of the obtained spectra, will give the readers a clear understanding of this article.

Reply:

We thank the reviewer for this helpful suggestion. Following the reviewers suggestion, we have added more experimental spectra to the revised SI (Figure S18 – S29). The impression of an apparent split of the cross peak at one specific waiting time for one sample is due to a combination of the chosen interval of the contour plots and the experimental noise at longer waiting times (see also our reply to the third point of the reviewer).

In case of two disparate peaks, the CLS analysis would still be possible. Two disparate peaks are just the limiting case of an inhomogeneously broadened line shape, for which – in the absence of exchange – the CLS would be constant at 1. Our center-line-slopes for the COO^- band, however, decay to 0. As such, our data indicate that exchange (being it chemical exchange of a bond between carboxylate and ion/water or exchange of the excited quanta) occurs within the experimentally accessible time scale of $\sim 1ps$.

Reviewer 2

Hunger and coworkers investigated the influence of salts on a peptide consisting of two alanine residues, 2Ala. Measurements were made in water with DR, NMR, IR, and 2D-IR. The author wanted to investigate conflicting conclusions as to whether cations or anions interact more strongly with peptides with the stated purpose of helping to "rationalize Hofmeister effects on proteins." Generally, I like the authors multi-spectroscopy approach and it is nice to see side-by-side data using DR, NMR, and vibrational spectroscopy on the same small molecule. I also agree with much of the local spectroscopic interpretations that the authors conclude. Despite this enthusiasm for the approach and data analysis, the authors have misinterpreted the meaning of some of their data. This will need to be corrected before this paper could be considered ready for publication. My detailed comments are as follows:

(1) The most important problem is the authors misconception that 2Ala contains "all fundamental protein motifs: C-terminus, amide backbone, and N-terminus." In fact, 2Ala is too small to stand in for the interactions of anions with a protein backbone. Most proteins are significantly longer, and the difference is crucial to understanding Hofmeister chemistry. The water structure around dipeptides contain fewer broken hydrogen bonds compared to larger polymers that have the same functional groups (ref. 59 as well as Thosar, Nat. Chem. 14, 2022, 8-10). Weakly hydrated anions like SCN⁻ bind more strongly at positions where the hydrogen bonding is broken.

Reply:

The reviewer seems concerned that 2Ala does not properly represent the amide backbone of a protein. We are somewhat surprised by the reviewer referring to our misconception, given that this statement seems to largely rely on a misquotation from our original submission. In our original submission we wrote:

"all fundamental protein motifs: C-terminus, amide bond, and N-terminus".

and

"2Ala carries all functional groups discussed above – both termini and an amide bond"

Hence, we did not refer to the amide backbone and our concept solely relies on the chemical motifs. As such, we agree with the reviewer that 2Ala cannot represent an amide backbone, which we deliberately did not refer to in our original submission. In the revised manuscript we have tried to clarify this by revising the statement, which now reads:

"carrying all fundamental chemical protein motifs: C-terminus, amide bond, and N-terminus".

As pointed out by the reviewer, also the interaction with water plays an important role for the interaction of macromolecules with ions. This has indeed been recently suggested from studies on solutions of

oligo-ethers (ref 59 of our original submission, ref 21 of the revised manuscript, and Thosar et al), which are also very dissimilar to a real protein. Nevertheless, as also pointed out in our response to reviewer 1, the use of macromolecules imposes some experimental challenges, as the change of an experimental observable due to interaction of an ion with the macromolecule cannot be unambiguously isolated from a variation of this observable due to a salt-induced change of the conformation. While 2Ala is certainly too short to fully represent a protein, its conformation is rather rigid. This rigidity allows us studying interaction of ions in the absence of conformational changes, which is a prerequisite for studying the competition between the different interaction sites at 2Ala. Elucidating this competing interactions is the scope of the present work.

To clarify the experimental challenges with conformationally flexible protein models, we discuss these in a separate paragraph in the introduction of the revised manuscript, which reads:

Above conclusions have been drawn using a variety of model compounds including simple molecules like N-methyl-acetamide,^{14,36-38} short oligo-peptides,^{8,39,40} and polymers.^{21,24,38,41,42} Comparison of the results using small molecules to those obtained with macromolecules^{21,38} has suggested that the length of the macromolecule largely impacts the interaction with ions. This different behavior has been explained by the different hydration structure of small molecules and macromolecules.²¹ Conversely, using large, conformationally flexible molecules makes it more challenging to isolate ion interactions: an ion-induced change of an experimental observable may stem from direct interaction of an ion with the model system or from an indirect, ion-induced change in the conformation.⁸ For instance, hydration of a salt may lead to dehydration and conformational variation of a macromolecule.⁴³

We further emphasize the relevance of hydration on specific ion effects in the first paragraph of the revised introduction, which reads:

Moreover, not only the chemical nature of the protein sites but also their hydration – intimately connected to the protein structure – affects interaction with ions.²¹

and clearly state that an amide bond does not fully represent the amide backbone of a protein:

Although 2Ala contains only one amide bond, and thus cannot fully represent the amide backbone of proteins, 2Ala has a rather rigid conformation,⁵¹ which limits changes in the spectroscopic observables due to conformational changes and thus allows isolating ionic interactions spectroscopically.

(2) The authors found that the 2Ala amide NH is very sensitive to the presence of KI and KSCN. They therefore concluded that the abundance of these sites on proteins would be the key reason that denaturing anions are more efficient than cations. For longer chain polymers, however, previous work has shown that when the NH group on an amide is alkylated, that denaturing anions still bind to roughly the same extent as when the NH group is present (Rembert, Langmuir, 31, 2015, 3459-3464). One possibility is that the NH group might interact even more strongly with the I⁻ or SCN⁻ if the chain were simply longer. Another is that broken hydrogen bonding might lead to

better interactions with the two methyl groups or the alpha carbon on 2Ala.

Reply:

We thank the reviewer for the comment. Indeed, our results suggest preferential interaction of anions with the amide group. As we have not performed experiments using a model compound with the N-H group methylated, our data do not provide any insights into the role of methylation. We agree that hydration can potentially play a role, yet the abundance of N-H groups in a protein may also play a role. However, we did not conclude this being the ‘key reason’ in our original submission, as perceived by the reviewer. We have revised this statement accordingly such that the abundance of amide groups can be one of the reasons for anions being stronger denaturants, which reads:

More general, the herein found preference of anions to the amide group and the preference for cations to the carboxylate groups may help rationalizing why denaturing anions are often more efficient denaturants than cations: amide groups are typically much more abundant in a protein than carboxylate groups.

(3) The data in Figures 2 and 4 are supposed to assure the reader that 250 mM 2Ala is a sufficiently low concentration that interactions between dipeptides can be ruled out in the subsequent work with salts in Figures 3, 5, and the infrared spectra. However, just because these peptides do not interact with each other in the absence of salt, does not mean that molar quantities of salt are not causing the dipeptide to associate with itself. The best way to test this would be to spot check the DR and NMR data with a lower concentration of 2Ala. This should be done down as close to the detection limit as possible.

Reply:

We agree with the reviewer that the absence of aggregation in the absence of salt does not necessarily exclude aggregation in the presence of salt. To test whether aggregation may affect our findings in the presence of salts, we performed H-NMR measurements for 50 mM 2Ala with varying concentration of KI solutions. We used KI as this salt showed the largest change in relative shift of the NH proton. These results at 50 mM 2Ala quantitatively confirm our observations for solutions at 250 mM 2Ala (see Figure R3).

Figure R3: Concentration-dependent relative chemical shift for aqueous solutions of a) 250 mM 2Ala and b) 50 mM 2Ala with increasing concentration of KI.

As for the DR experiments, it is more challenging to confirm our findings at lower concentrations of 2Ala. While it is possible to isolate the contribution of 2Ala at low concentrations in the absence of salt, where the conductivity of the samples is negligible, the spectral decomposition is not accurate enough at high salt concentrations. The reason for this can be found in the high electrical conductivity of the samples, which impose an additional Ohmic loss term. This Ohmic loss term scales with the samples conductivity and the inverse frequency. Experimental noise is approximately proportional to the Ohmic loss and main source of the experimental uncertainty at low frequencies.

We have recorded dielectric spectra at 125 mM 2Ala, yet all ion-specific variations of the 2Ala relaxation are almost within experimental error. This can be explained by the above considerations: the experimental errors at 125 and 250 mM are comparable (as the samples have nearly the same conductivity), but the relaxation strength of 2Ala is two times lower. The inability to detect ion-specific changes is also consistent with our data in Figure 3c: At a two times reduced concentration of 2Ala, the relaxation strength is reduced to ~ 10 . We observe an ion-specific change of the relaxation strength of up to $\sim 20\%$. Hence, the expected ion-specific variation at 125 mM 2Ala of ~ 2 is close to the experimental error of ± 0.5 . As such DR experiment at 125 mM are barely conclusive.

To test the effect of the 2Ala concentration on the IR spectra, we have additionally recorded infrared absorption spectra of 2Ala in the presence of varying concentrations of LiCl and tested the effect of the concentration of 2Ala. As can be seen in Figure R1, the salt induced changes to the amide I and carboxylate band at 50 mM 2Ala coincide with those determined at 250 mM of 2Ala. As such, also our conclusions based on the IR spectra are not affected by the concentration of 2Ala.

We have included these additional NMR and IR experiments in the SI of the revised manuscript (Figure S11 and S14) and discuss the effect of 2Ala concentration in the revised main text.

In summary, this paper is potentially interesting because using 4 different spectroscopies to study the same small molecule can potentially help the Hofmeister community gain a better appreciation of the differences in information that each technique can provide

about ion interactions with dipeptides. However, the authors stated that they wanted to resolve the “orthogonal conclusions” that “have been obtained with either cations or anions interacting more strongly with model peptides.” It would be best for the authors to simply acknowledge that NMA (ref. 14) and 2Ala are too short to provide proper information on the interactions of denaturing anions with the polypeptide backbone of full length proteins.

Reply:

We are again somewhat surprised about this statement. The present manuscript does not use NMA and, indeed, we set out for resolving seemingly orthogonal conclusions for model peptides. We are not aware of any reason that would prevent classifying 2Ala as a model peptide.

We acknowledge that 2Ala is not a protein, yet it is unclear to us why the reviewer got the impression that our findings can be directly related to proteins. Except for the introduction, we only refer to proteins in the context of ‘salts with increasing/decreasing protein denaturation tendency’ in the main text. Moreover, although studies of longer peptides have provided valuable insights into the origins of specific ion effect, their conformational flexibility impose additional experimental challenges, which we now explain in the revised introduction

Reviewer 3

In this manuscript, the authors utilize a trio of spectroscopies to answer the age old question of the salt effects on the peptide backbone. It has been brought into question several times during discussion on folding and dynamics. The authors use 2D IR spectroscopy to directly detect any changes on the fast dynamics of the changing electric fields around both the amide I CO and the C-terminal COO to characterize the weak interactions with different salt concentration. The first two spectroscopies highlight both those denaturing cations and anions have significant effects to different areas of the peptide. Using 2D IR, the authors directly observe and measure that the salt has minimal effect on the vibrational relaxation and the correlation decay. This hints that the fast dynamics remain virtually unaffected. However, one exception is Li⁺ ions. These ions interaction strongly with the terminal COO in a concentration dependent way. These observations provide insights into microenvironments and preferential solvation responsible for weak interaction. The overall insights of this paper are both interesting and significant to contribute to field of peptide studies and thus the paper should be accepted upon addressing the following concerns and revisions:

a) Although spectral shifts are not always detectable, the reviewer wonders how different the bandwidth is in the linear IR measurements since the bandwidth has contributions of CLS and vibrational relaxation.

We thank the reviewer for the comment. To determine the linewidths of both bands in the linear IR spectra, we fitted the COO⁻ and CO mode with a Gaussian line shape (see Figure R4 a). Within the experimental error (resolution of the linear IR measurement, 4 cm⁻¹, see experimental section), no significant changes in FWHM (full width at half maximum) can be detected, except for the increasing linewidth of the COO⁻ mode for the case of LiCl. This observation is in line with the increasing inhomogeneity of the COO⁻ with increasing concentration of LiCl. As such, the observed changes in linewidth for the COO⁻ mode for LiCl are consistent with our 2D IR results.

We have added this analysis to the revised SI and figure R4 is now included as figure S13.

Figure R4: a) Linear IR spectrum of 2Ala fitted with two Gaussian functions: Red for the COO⁻ mode at 1590 cm^{-1} and blue for the CO mode at 1660 cm^{-1} . Salt and concentration dependent FWHM of COO⁻ mode (b) and CO mode (c). Error bars stem from the resolution of the linear IR measurements (4 cm^{-1}).

b) In line 326-327, the authors mention energy transfer taking place, yet the authors do not address any influence of the Li⁺ ions on the transfer rates. In other words, does the relaxation channel opened by the coupling increase with Li⁺ concentration? If not, why?

Reply:

The reviewer raises an excellent point. We did not detect any differences in the cross peak dynamics for 2Ala and 2Ala + 5 M LiCl solutions. To better illustrate these similar dynamics, we show the evolution of the diagonal and cross peak integrals in Figure R5 (Figure S16 of the revised submission). This figure demonstrates that both, the dynamics and the magnitude (relative to the diagonal peak), of the cross-peak are very similar for 2Ala and 2Ala+5 M LiCl. Based on solely the experimental data, we unfortunately cannot pinpoint the reason why the presence of Li⁺ does not affect the relaxation channel, despite the fact that addition of Li⁺ increases the overlap of the amide I band and the carboxylate band. The coupling strength, which determines the energy transfer efficiency, likely depends on the frequency of the coupled oscillators, the magnitude of their transition dipole moments (for electrical coupling), and also the relative orientation of the coupled transition dipoles. We could speculate that coordination of Li⁺ to the carboxylate group in a mono-dentate manner lifts the degeneracy of both C-O groups of COO⁻ and thereby also affects frequency, the transition dipole, and the orientation of the transition dipole of the COO⁻ mode relative to the amide I mode. Apparently, these changes largely cancel for the coupling strength.

Figure R5: Integrated peak volumes of the marked areas of the 2D-IR spectrum of 2Ala (see inset) as a function of waiting time for 2Ala (solid lines) and 2Ala+5M LiCl (dotted line). The pink symbols show the volume of the bleaching signal at the diagonal, orange symbols the evolution of the cross peak.

We have added this analysis (Figure S16 of the revised SI) and the discussion in the supplementary information of the revised submission, which reads as follows:

In the main manuscript we show in Figure 7 b the appearance of a cross peak between the amide I and the COO⁻ mode at later waiting times. As can be seen from the integrated peak volumes shown in Figure S16, the cross-peak has its maximum intensity at a waiting time of ~200 fs, indicative of energy transfer from the amide I mode to the low frequency carboxylate mode. These dynamics are rather insensitive to the addition of 5 M LiCl. Also the intensity of the cross-peaks, relative to the signal intensity of the diagonal signal of the amide I mode for 2Ala in water and 2Ala + 5 M LiCl are virtually the same, which suggests that addition of LiCl hardly affects the coupling strength of both vibrations. As such, our data suggest that, despite LiCl alters the vibrational structure and dynamics of the COO⁻ mode (as discussed in the main manuscript), the coupling strength to the amide I mode remains unaffected. This may be due to a cancellation of altered coupling due to a simultaneous variation of the transition dipole strength, transition dipole orientation, and resonance frequency of the COO⁻ mode upon interaction with Li⁺.

c) The dynamics of folding and activity would most likely be influenced by amide I interactions. The authors should discuss what influence if any that interaction with the termini would cause. How does this apply to the bigger picture?

Reply:

We thank the reviewer for this insightful point. Indeed, interaction of Li⁺ with the charged termini has presumably only a limited effect on the amide backbone of a real protein. Yet, it may affect stabilizing

motifs that rely on charged residues (e.g. so called salt-bridges). Moreover, our results based on the short peptide 2Ala may be of relevance to salt effects on the self-assembly of short peptides, where interaction between oppositely charged sites can play a pivotal role. We have added these points to the conclusions of the revised manuscript, which reads:

This interaction with the carboxylate presumably has only limited effect on the backbone of a real protein, but may of relevance to the interactions of charged residues that stabilize proteins.⁸⁷ Also for the self-assembly of short peptides charged sites can play a key role,⁸⁸ and our results suggest that LiCl can efficiently distort such interactions.

d) The author uncover from the 2D IR measurement that Li⁺ is the ion having the greatest interaction with the termini. What is special about the Li⁺ ions that make the interaction possible?

Reply:

The reviewer raises an interesting question. The marked effect of LiCl on the vibrational structure of the COO⁻ mode is consistent with the general tendency of Li⁺ having the highest ion-pairing tendency with carboxylate salts amongst alkali metal cations. Interestingly, recent studies (see e.g. van der Vegt et al. Chem. Rev. 2016, 116, 13, 7626–7641 for an overview) have suggested that this is not due to enhanced direct contacts between Li⁺ and the carboxylate, but rather due to carboxylate-Li⁺ contacts separated by at least one water layer. We refer to these studies in the discussion section of the revised version of the manuscript, which reads:

Together, these observations suggest that LiCl markedly alters the microenvironments and their dynamics for the COO⁻ group: Li⁺ cations interact with the negatively charged carboxylate group. The enhanced tendency of Li⁺ to interact with 2Ala's carboxylate group is in line with the general tendency of alkali cations to bind to carboxylates.⁸⁶ In this context it is interesting to note that for the interaction of Li⁺ with carboxylates direct ion contacts have been suggested to play only a minor role, rather solvent-separated contacts prevail.⁸⁶ Our observation that LiCl can efficiently alter the vibrational structure of 2Ala's carboxylate group may thus only partially stem from direct contacts. Yet, also long-lived contacts with hydrated ions could alter 2Ala's vibrational dynamics, as water in ionic hydration shells is less dynamic than in bulk water.

REVIEWERS' COMMENTS:

Reviewer #1 (Remarks to the Author):

The authors responded to most of the comments raised by the reviewer. The revised manuscript is in a much better format than before. The reviewer believes that this work is publishable in Communications Chemistry.

Reviewer #2 (Remarks to the Author):

The statement the authors make in the abstract "carrying all the fundamental chemical protein motifs: C-terminus, amide bond, and N-terminus" is technically true. Nevertheless, it is misleading. The authors state in the same sentence that they are doing this work "to provide a comprehensive picture." Moreover, in the sentence before this they talk about "our understanding of Hofmeister effects on proteins" and "conflicting conclusions." Also, the title states "Cations Bind to C-terminus, Anions to the Amide" is technically true for alanylalanine but may also mislead readers into believing the results here extrapolate to c-termini and amide groups on proteins.

The additional paragraph the authors have put in the introduction is a start, but they need to be clearer that 2Ala is not going to provide a comprehensive picture for the interactions of ions with proteins. The problem with their statements is that the authors are overselling their case. They have done very nice work on 2Ala, but this small molecule may not be a good model for the C-terminus, the amide bond, or the N-terminus of proteins. This dipeptide is simply too small for that. They need to tone down the language that they are using. Moreover, the "conflict" to which they are referring, in part, includes the authors' own previous mistakes. Specifically, they stated in ref. 14 in *Angew. Chemie* (2016), "Comparing the present findings with results for cations shows that in contrast to common belief, anion–amide binding is weaker than cation–amide binding." The mistake in their previous paper came from extrapolating from work on NMA, which is another small molecule that also may not have much to do with ion interactions with proteins. The author should acknowledge here that the conclusions they are now drawing are different from the ones they drew earlier. Also, they should more directly state that 2Ala is not a protein.

In conclusion, I like this paper and would be glad to see it published in Communications Chemistry. It would be better, however, if this were done with less hype.

Reviewer #3 (Remarks to the Author):

The authors have addressed my concerns and questions. The paper is probably publishable if the authors have sufficiently answered the other reviews.

Reviewer 1

The authors responded to most of the comments raised by the reviewer. The revised manuscript is in a much better format than before. The reviewer believes that this work is publishable in Communications Chemistry.

Response:

With thank the reviewer for carefully reading our response and are pleased that the reviewer is satisfied with the revisions.

Reviewer 2

The statement the authors make in the abstract "carrying all the fundamental chemical protein motifs: C-terminus, amide bond, and N-terminus" is technically true. Nevertheless, it is misleading. The authors state in the same sentence that they are doing this work "to provide a comprehensive picture." Moreover, in the sentence before this they talk about "our understanding of Hofmeister effects on proteins" and "conflicting conclusions." Also, the title states "Cations Bind to C-terminus, Anions to the Amide" is technically true for alanylalanine but may also mislead readers into believing the results here extrapolate to c-termini and amide groups on proteins.

Response:

We have carefully read our manuscript and tried to identify statements that may have mislead the reviewer such that the reviewer gets the impression that our results can be extrapolated to proteins. Although we are not fully aware which statements may have resulted in the reviewer's perception, we have tried to rephrase the title and the abstract. Our original title clearly stated that our study investigates "Specific Ion Effects on Alanylalanine". In order to meet the format requirements of Communications Chemistry, we have revised the title as follows:

"Ion-specific binding of cations to the carboxylate and of anions to the amide of alanylalanine"

We hope that this title is less prone to miss-interpretations.

We have also tried to emphasize in the abstract (within the limitation of the abstract to 200 words), that 2Ala is a simple, small model peptide, which reads:

"..., we study a small model peptide, L-Alanyl-L-alanine (2Ala), carrying all fundamental chemical protein motifs: C-terminus, amide bond, and N-terminus."

The additional paragraph the authors have put in the introduction is a start, but they need to be clearer that 2Ala is not going to provide a comprehensive picture for the interactions of ions with proteins.

The problem with their statements is that the authors are overselling their case. They have done very nice work on 2Ala, but this small molecule may not be a good model for the C-terminus, the amide bond, or the N-terminus of proteins. This dipeptide is simply too small for that. They need to tone down the language that they are using.

Response:

We have carefully read the manuscript and tried to identify instances, which may result in the perception that 2Ala can be interpreted to represent a full protein. In the revised manuscript we now explicitly state on two occasions in the introduction that 2Ala cannot fully represent a protein, which reads:

“Comparison of the results using small molecules to those obtained with macromolecules^{21,38} has suggested that small model systems cannot account for all details of ion-specific effects on proteins and the length of the macromolecule largely impacts the interaction with ions.”

and

“Although 2Ala contains only one amide bond, and thus cannot fully represent the amide backbone of proteins, 2Ala has a rather rigid conformation,⁵¹ which limits changes in the spectroscopic observables due to conformational changes and thus allows isolating ionic interactions spectroscopically.”

We have further revised the beginning of the “results and discussion section”, which reads:

“To provide a consistent view on ion-specific effects on a small model peptide”

Moreover, the “conflict” to which they are referring, in part, includes the authors’ own previous mistakes. Specifically, they stated in ref. 14 in *Angew. Chemie* (2016), “Comparing the present findings with results for cations shows that in contrast to common belief, anion-amide binding is weaker than cation-amide binding.” The mistake in their previous paper came from extrapolating from work on NMA, which is another small molecule that also may not have much to do with ion interactions with proteins. The author should acknowledge here that the conclusions they are now drawing are different from the ones they drew earlier. Also, they should more directly state that 2Ala is not a protein.

Response:

We acknowledge that the observations here are different than for NMA, but disagree with the reviewer that our earlier conclusion were a “mistake”. Similar to 2Ala being dissimilar to a protein, 2Ala is also dissimilar to NMA: Hence our conclusions from ref 14 are still valid: Our results for the isolated NMA molecule suggested that cations can perturb NMA more efficiently than anions. However, in the presence of the carboxylate group for 2Ala, our present results suggest that cations prefer to interact with the carboxylate group. To emphasize this difference, we have revised the according sentence of the conclusions, which now reads:

*“Although previous reports have indicated that salts like Li^+ can bind to the amide group,^{18,31} and that for an isolated amide group in *N*-methylacetamide cations can perturb the amide more efficiently than*

anions,¹³ our results show that in the presence of an amide group and a C-terminal COO⁻ group, the carboxylate group is the prevailing interaction site for Li⁺.”

In conclusion, I like this paper and would be glad to see it published in Communications Chemistry. It would be better, however, if this were done with less hype.

Response:

We hope that these revisions make the reviewer to perceive the manuscript with less hype.

Reviewer 3

The authors have addressed my concerns and questions. The paper is probably publishable if the authors have sufficiently answered the other reviews.

Response:

We are pleased that the reviewer feels, that his concerns were adequately addressed.